# Characteristics of VOCs Released from Plywood in Airtight Environments

**Tianyu Cao [1], Jun Shen [1],\*, Qifan Wang [1], Huifang Li [1], Cong Xu [1] and Huajun Dong [1,2]**

[1] College of Material Science and Engineering, Northeast Forestry University, Harbin 150040, China
[2] College of Academy of Design and Art, Harbin University of Commerce, Harbin 150040, China
\* Correspondence: shenjun@nefu.edu.cn; Tel.: +86-13936442508

**Abstract:** In order to explore the emission characteristics of volatile organic compounds (VOCs) and different VOC components in airtight environments, polyvinyl chloride laminated plywood (PVC-P), melamine-impregnated paper laminated plywood (MI-P), water-based paint laminated plywood (WP-P) and unfinished plywood (UF-P) were tested as materials in 15 L small environment cabins. VOCs were collected after being sealed for 2 h, 4 h, 6 h, 8 h, 12 h, 18 h, 24 h and 30 h under different loading ratios (1 $m^2/m^3$, 1.5 $m^2/m^3$, 2 $m^2/m^3$, 2.5 $m^2/m^3$) and analyzed using a gas chromatography–mass spectrometer. The results show that VOCs gradually increased and tended to be saturated with the increase of time in an airtight environment. The saturation of veneered plywood was faster than unfinished plywood and the pollution degree of three kinds of veneered plywood from lowest to highest was in order of PVC-P, MI-P, WP-P. Aromatic hydrocarbons account for more than half of VOC concentration, and was most obviously affected by the loading ratios. Surfaced laminated plywood can reduce the emission of aromatic hydrocarbons, but also increases the amount of other compounds released. The concentration of VOC-components shows different characteristic curves at different loading ratios due to the influence of decorative materials.

**Keywords:** plywood; volatile organic compounds; airtight environment; surfaced laminated materials; loading ratios

## 1. Introduction

In order to protect the natural environment, many countries and regions have implemented policies restricting logging and use of wood-based panels such as plywood, leading to particleboard being used more and more widely [1]. Plywood is widely used in furniture manufacturing because of its advantages such as its resistance to deforming, good transverse tensile properties and large breadth. However, plywood also brings harm in the process of its manufacture and use which releases volatile organic compounds causing indoor air pollution. In modern society, people yearn to live in the city where life is more convenient. However, air pollution in city centers is worse than in the countryside, and can cause a series of physical discomforts [2,3]. Highly toxic substances such as formaldehyde and benzene can irritate eyes and mucosa, resulting in sick building syndrome (SBS) and symptoms such as headaches, fatigue and nausea [4]. VOCs are especially harmful to the elderly and children, and can lead to cancer and even death [5,6]. Therefore, more and more attention has been paid to research on VOCs.

In recent years, there have been studies on the impact of volatile organic compounds (VOCs) on indoor air and health risk assessments. The total concentration and composition of VOCs were analyzed. The comprehensive index is calculated according to the various guidance limits of VOCs proposed by Bernd, and then the indoor air pollution level is determined [7,8]. The odor of panel furniture has been analyzed in some studies and the results show that the concentration and odor of

total volatile organic compounds (TVOCs) will increase with the increase of ambient temperature and relative humidity [9]. VOCs released from wood itself, surface coatings and decorative materials can also be a source of odor [10,11]. Although the VOC release in wood-based panels can be blocked and the odor reduced, some VOCs could also be released from the wood-based panel itself, which can cause an increase in the VOCs released within a short timeframe and make the odor worse [12]. Aromatic compounds and esters are the main sources of odor in melamine veneered particleboard [13,14]. PVC is a thermoplastic polymer made from vinyl chloride monomers. The VOCs released by PVC mainly includes dibutyl phthalate and dioctyl phthalate [15]. It is easy to release vinyl chloride, trichloromethane and trichloroethylene when PVC is heated, whereby it produces an irritating acidic plastic smell [16].

Environmental parameters, production processes and finishing materials are important factors affecting the emissions of VOCs [17]. Research shows that an increase in temperature can promote the release of formaldehyde and VOCs in wood-based panels, more so than an increase in relative humidity [18,19]. The release rate of VOCs increases by increasing hot-pressing temperature and time [20]. Most wood-based panels were decorated with veneers and edges in order to make them look beautiful and reduce air pollution [21]. Research shows that different veneers have different sealing rates for formaldehyde and TVOCs [22,23]. Using polyethylene film veneers and pine veneers, formaldehyde emission of wood-based panels has been reduced [24]. The release of TVOCs and VOC components are affected by decorative treatments [25]. Veneer treatments and surface painting can effectively reduce TVOC release from particleboard, but surface painting and wood modification with chemical reagents often lead to increased TVOC release, which brings hidden dangers to human health [26,27].

In this experiment, the VOC emission of four kinds of plywood are measured where the air exchange rate was 0 times/h. Four loading ratios (1 $m^2/m^3$, 1.5 $m^2/m^3$, 2 $m^2/m^3$, 2.5 $m^2/m^3$) and eight airtight storage periods (2 h, 4 h, 6 h, 8 h, 12 h, 18 h, 24 h and 30 h) were set. The total concentration and composition of VOCs were qualitatively and quantitatively analyzed by using a gas chromatography–mass spectrometer (GC-MS). The characteristic curve of VOC released by plywood under closed conditions was explored, and the effects of finishing materials and loading ratios on the release of VOCs from plywood were analyzed. It will be of great guiding significance when choosing plywood for furniture decoration.

## 2. Materials and Methods

### 2.1. Materials

In this experiment, the four kinds of plywood used were E1 grade (the formaldehyde emission level) and from a furniture manufacturer in Guangzhou. Polyvinyl chloride laminated plywood (PVC-P), melamine-impregnated paper laminated plywood (MI-P) and unfinished plywood (UF-P) were ready-made, but the water-based paint laminated plywood (WP-P) was hot-pressed and hand painted before the experiment. The length × width × thickness used was 1200 mm × 1200 mm × 8 mm, and the pH was 7.2–7.4. The hot-pressing temperature was 190–200 °C and hot-pressing time was 210 s. The initial moisture content of plywood ranged from 8.5% to 10.5%. The four kinds of plywood were cut to 150 mm × 50 mm × 8 mm, and 150 mm × 75 mm × 8 mm. Aluminum foil was used to seal the edges of the specimens so that the VOCs do not escape from the edges, then stored at –30 °C.

### 2.2. Equipment

Specifications of the equipment used are as follows:

(1) A 15 L small environment cabin was used as the VOCs sampling chamber, which was made by Northeast Forestry University. A 15 L small environment cabin has the advantages of simple assembly, low cost and a good correlation with a 1 $m^3$ environment cabin. Nitrogen (99.99% purity) was used as carrier gas, at a temperature of 23.5 °C ± 0.5 °C and a gas exchange rate of 0 $h^{-1}$.

(2) Tenax-TA tubes (L × R = 89 mm × 3.2 mm, with 200 mg of fillers inside to absorb VOCs including n-hexane to n-hexadecane, BeifenTianpu Instrument Technology Co., Led. Beijing, China) were used to collect the gas released from plywood in the cabin.

(3) A vacuum sealing machine (VS2110GB, Dongguan Yinger Electrical Appliances Co., Ltd., Dongguan, Guangdong Province, China) was used to vacuum samples into polytetrafluoroethylene bags.

(4) An Analytical Tube Processor (TP-2040, Beijing BeifenTianpu Instrument Technology Co., Ltd., Beijing, China) was used for thermal desorption of the Tenax-TA tubes and removal of residues in the tubes.

(5) A miniature vacuum pump (ANJ6513-220V, Chengdu Xinweicheng Technology Co., Ltd., Chengdu, Sichuan Province, China) was used as a piece of sampling equipment. A Tenax-TA tube was set between the 15 L small cabin and vacuum pump, and gas was collected into the tube by vacuum extraction.

(6) A Thermal Desorber (ultra&unity, Markes International Inc., Llantrisant, UK) was used for thermal desorption. The cold trap adsorption temperature was −15 °C, analytical temperature was 300 °C and pipeline temperature was 180 °C.

(7) A GC-MS (DSQII, Thermo Scientific, Waltham, Massachusetts, U.S.A) was used to characterize and quantify the VOCs. The basic parameters were as follows: the type of chromatographic column was DB-5MS, 30 m × 0.25 mm × 0.25 μm; the carrier gas was helium at 99.996% purity; the injection port temperature was 250 °C; and the distribution ratio was 40. The temperature program had three steps: first kept at 40 °C for 2 min, then increased to 150 °C by 4 °C/min and kept for 4 min, and finally increased to 250 °C by 10 °C/min and kept for 8 min. For electron ionization: 70 eV were used, the ionization temperature was 230 °C, transmission line temperature was 250 °C and the mass scan range was 40–450 amu with a full scan.

### 2.3. Methods

#### 2.3.1. Sampling

The samples were divided into four groups: PVC-P, MI-P, WP-P and UF-P, with two pieces measuring 150 mm × 75 mm × 8 mm and four pieces measuring 150 mm × 50 mm × 8 mm according to the loading ratios and exposure area of the panel (1 $m^2/m^3$-0.015 $m^2$, 1.5 $m^2/m^3$-0.225 $m^2$, 2 $m^2/m^3$-0.03 $m^2$, 2.5 $m^2/m^3$-0.0375 $m^2$).

The interior walls of the 15 L small cabin was cleaned by anhydrous ethanol and distilled water. Then the fan was turned on and kept on for more than 30 min and nitrogen with 99.99% purity was injected. The main experimental parameters were temperature at 23.5 °C ± 0.5 °C controlled by air conditioner, and a gas exchange rate of zero. Samples were then placed into the cabin, and kept in an airtight condition by using the fan and nitrogen.

Before collection, the Tenax-TA tubes should be desorbed by using the analytic tube processor for 30 min at 325 °C. A miniature vacuum pump was used to collect 3 L of gas in the cabin. In this way, eight experiments were conducted in each group and the vacuum times were 2 h, 4 h, 6 h, 8 h, 12 h, 18 h, 24 h and 30 h, respectively.

#### 2.3.2. Analytical Methods

Toluene-D8 was used as the solute and dissolved in methanol to make a standard curve. 2 μL of Toluene-D8 was injected into a Tenax-TA tube and pre-purged for 5 min. The DSQII gas chromatography–mass spectrometer was used to characterize and quantify the VOCs. In the analysis of VOCs, firstly the volatile components with a greater than 90% match and the number of carbon atoms between 6-16 were selected, then the volatile components were analyzed by retention time. Finally, the peak area of $C_7D_8$ was used to quantify VOC concentration.

## 3. Results and Discussion

### 3.1. Comparison of Dynamic and Airtight Experiment

PVC-P, MI-P, WP-P and UF-P were selected for 4 h dynamic (gas change rate of 1 h$^{-1}$) and 4 h airtight (gas change rate of 0 h$^{-1}$) experiments.

Figure 1 shows that the concentration of VOCs in dynamic experiments was lower than that in airtight experiments. The emission characteristics of VOCs were influenced by the laminated surface and the differing loading ratios in the dynamic and airtight experiments. The emission characteristics of VOCs from laminated plywood could be simulated by dynamic experiments under ideal ventilation conditions, while the airtight experiments could simulate characteristics of VOCs of an enclosed living environment under extreme conditions. Both experiments have guiding significance for the selection of panel materials in everyday settings under different ventilation conditions. At present, there are many pieces of research on VOCs released in dynamic environments. Therefore, it is necessary to study the characteristics of VOCs released in airtight environments.

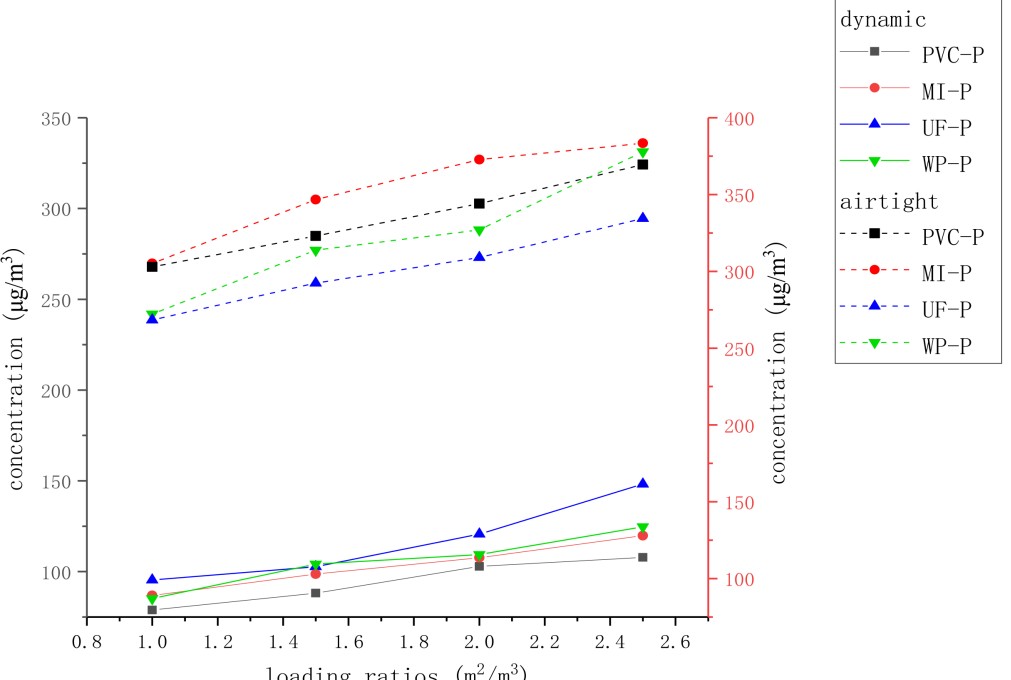

**Figure 1.** Comparison of 4 h dynamic and 4 h airtight experiments.

### 3.2. Trend of VOCs of Plywood with Time under Four Loading Ratios

Figure 2 shows the concentration of the VOCs released from PVC-P, MI-P, WP-P and UF-P after storage for 2 h, 4 h, 6 h, 8 h, 12 h, 18 h, 24 h, and 30 h respectively in an airtight environment cabin. Regardless of the plywood type, VOCs concentration increased with time stored in an airtight environment, and gradually reached saturation; however, the speed of saturation of plywood with different laminate materials was different. The concentration of VOCs of PVC-P was higher by 97.15%, MI-P by 127.27%, and WP-P by 152.71% after storage for 12 h compared with storage for 2 h. However, the concentration after storage for 30 h only increased by 3.22% (PVC-P), 2.5% (MI-P) and 2.76% (WP-P) from storage for 12 h. Therefore, the concentration of VOCs released from veneered plywood (PVC-P, MI-P, WP-P) reached saturation faster, after around 12 h. However, the VOC concentration of UF-P needs 18 h to reach the saturation, and increased only 0.66% from 18 h to 30 h.

UF-P only releases VOCs from the interior plywood, but laminated materials release VOCs from the inner board as well as the veneer, as the covered surface itself contains VOCs. Taking WP-P as

an example, the water-based paint coated on it contains a large amount of volatile organic compounds from the frontal benzene series and will release a large number of volatile organic compounds quickly in a short time. Therefore, when the storage time in an airtight environment for WP-P was short (2 h–8 h), the concentration of VOCs increased fastest and reached its peak saturation. In terms of VOC concentration at saturation combined with the pollution degree of the three types of veneered plywood, PVC-P was best, MI-P followed and WP-P was the worst.

Plywood is a kind of wood-based panel, and the pore size distribution of wood-based panels is wide. Fick diffusion, Kundsen diffusion and transition diffusion coexist in the plywood [28]. The diffusion of VOCs is affected by the loading ratios and the concentration of the air-board in the two-phase interphase in the environment cabin. The higher the loading ratio is, the higher the VOCs concentration released from the plywood is. WP-P was least affected by the loading ratios, with the VOCs concentration at 2.5 m$^2$/m$^3$ only 1.12 times that at 1 m$^2$/m$^3$.

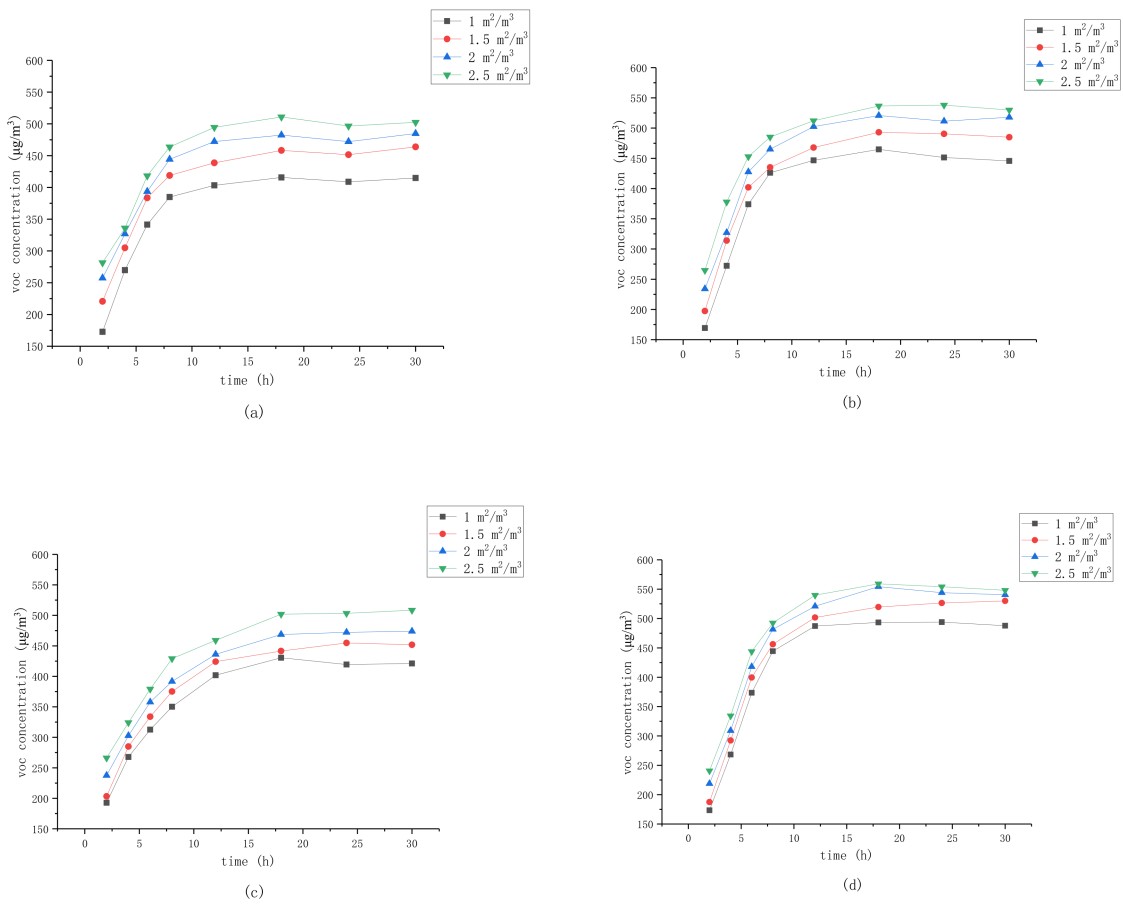

**Figure 2.** Trend of VOCs with airtight time under different loading ratios: PVC-P (**a**), MI-P (**b**), UF-P (**c**) and WP-P (**d**).

### 3.3. Effect of Finishing Materials on VOC-Components Proportion

VOCs are aggregates of many volatile organic compounds; they are classified into eight categories according to their composition (aromatics, alkanes, olefins, aldehydes, ketones, alcohols, esters and others). Among them, aromatic hydrocarbons, aldehydes and esters have obvious odors, and aromatic hydrocarbons and aldehydes do great harm to human body. Figure 3 shows the proportion of VOCs released by PVC-P, MI-P, UF-P and WP-P after 30 h of airtight storage which were averaged under four loading ratios. Aromatic hydrocarbons account for the largest proportion of VOCs, which is consistent with the experimental results of Park et al. in Korea [29], because the components with higher release rate can better represent the release characteristics of VOCs [30]. Relevant studies have

shown that benzene series and other substances account for a large proportion of VOCs in closed air [31]. The proportion of aromatic hydrocarbons in VOCs released by four kinds of plywood was 65.63% (PVC-P), 68.34% (MI-P), 67.98% (UF-P) and 71.5% (WP-P), respectively.

It shows that veneer materials such as PVC and melamine impregnated paper can prevent the release of aromatic hydrocarbons from the inside of the plywood, while water-based paint itself contains a large number of aromatic hydrocarbons, as the aromatic hydrocarbons released from WP-P account for the largest proportion. Aldehydes and ketones accounted for the largest proportion of UF-P. Aldehydes accounted for about 6% and ketones accounted for 3.26%, which was 2–3 times as much as other kinds of plywood. The proportion of alkanes released from PVC-P is the largest, accounting for about 6.55%, which is higher than other veneer plywood and UF-P by 1–2 percentage points. The esters released from veneered plywood ranged from 11% to 12%, but only 9.62% from UF-P. This is due to the lack of decorative materials on the surface of veneer plain board, so a part of the source of esters is missing. Result shows research found that veneer treatments reduced the total amount of TVOCs released from sheets, but also increased the amount of certain compounds released [32], which also explains the increase in esters concentration released from MI-P and WP-P.

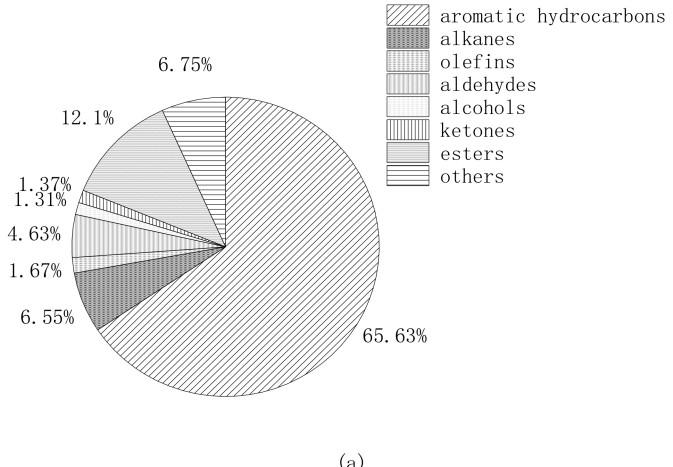

(a)

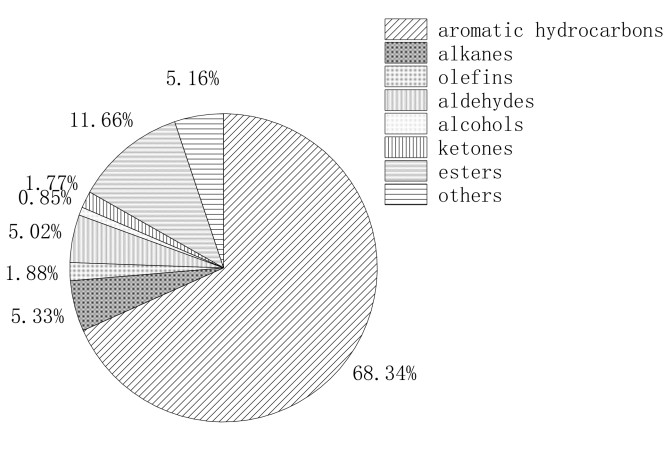

(b)

**Figure 3.** *Cont.*

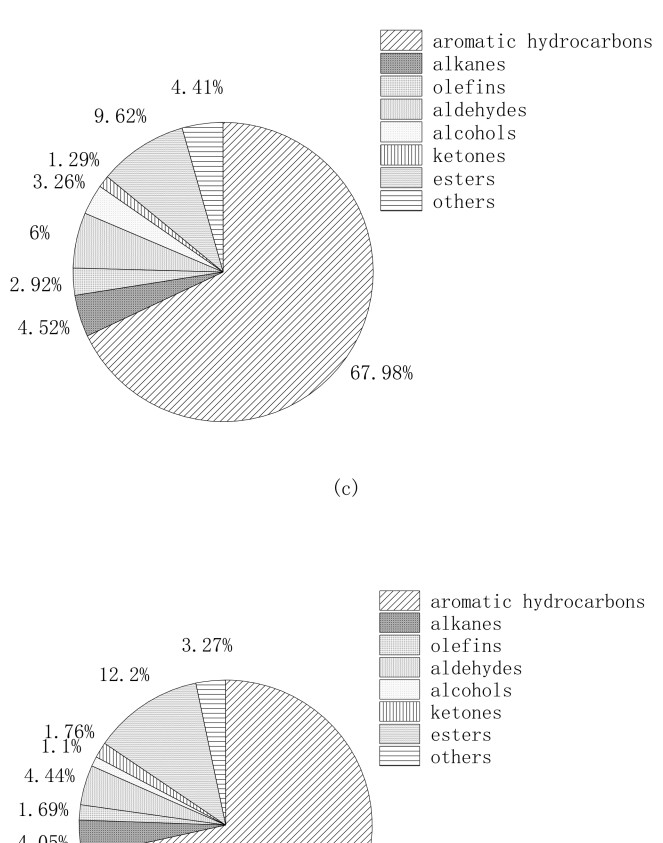

(c)

(d)

**Figure 3.** Proportion of VOCs released from PVC-P (**a**), MI-P (**b**), UF-P (**c**) and WP-P (**d**) at 30 h.

*3.4. Effect of Loading Ratios on VOC-Components Concentration*

Figure 4 shows the effect of loading ratios on VOC-components released from plywood in their saturated condition. As mentioned above, among the various components of VOC, the larger proportion is more representative, so the aromatic hydrocarbons with the largest proportion are more obviously affected by the loading ratios. Figure 4 shows that the effect of the loading ratios on WP-P was the smallest among the four kinds of plywood. When the loading ratios changed from 1 $m^2/m^3$ to 1.5 $m^2/m^3$, the concentration of aromatic hydrocarbons released from the WP-P increased by 4.02%; when loading ratios changed from 1.5 $m^2/m^3$ to 2 $m^2/m^3$ it increased by 0.62%; and by 5.37% when the loading ratios changed from 2 $m^2/m^3$ to 2.5 $m^2/m^3$. Overall, the concentration of aromatic hydrocarbons at 2.5 $m^2/m^3$ was 110.29% of that at 1 $m^2/m^3$. The concentration of aromatic hydrocarbons released from the other three kinds of plywood increased by more than 120%: PVC-P increased by 127.58%, MI-P by 132.89% and UF-P by 124.19%. There was also a consistent trend among PVC-P, MI-P and UF-P, when the loading ratio was lower, the change of aromatic hydrocarbon concentration was greater. For example, the aromatic hydrocarbons increased by 12%, 9.87% and 8% respectively in the process of increasing the loading ratios of MI-P, and 12.04%, 6.22% and 4.35% respectively in the case of UF-P. In addition to WP-P, the aromatic hydrocarbon concentration released by the other two types of veneered plywood were more affected by the loading ratios than that of UF-P.

The VOC-components released from the different types of laminated plywood, are affected by the loading ratios. The alkanes released from PVC-P and WP-P positively correlated with loading ratios.

When the airtight time was 30 h, the alkanes released from PVC-P increased by 5.65%, 22.04% and 8.52% with the loading ratios increasing step by step, while the alkanes released from WP-P increased by 23.09%, 6.56% and 3.49% respectively.

It can be seen that not all the release characteristics trend in a logarithmic growth as expected, but that it shows different characteristic curves because of the influence of decorative materials. Results show that the loading ratios do not have a linear relationship with VOC concentration [33]. The emission of VOCs from the inside plywood could be limited when the concentration increases to the maximum under the condition that air exchange rate is zero [34], which explains the different characteristic curves above. However, the obtained characteristics curve is limited because the loading ratios chosen in this experiment range only from 1 m$^2$/m$^3$ to 2.5 m$^2$/m$^3$. Therefore, it is necessary to carry out follow-up experiments with loading ratios greater than 2.5 m$^2$/m$^3$.

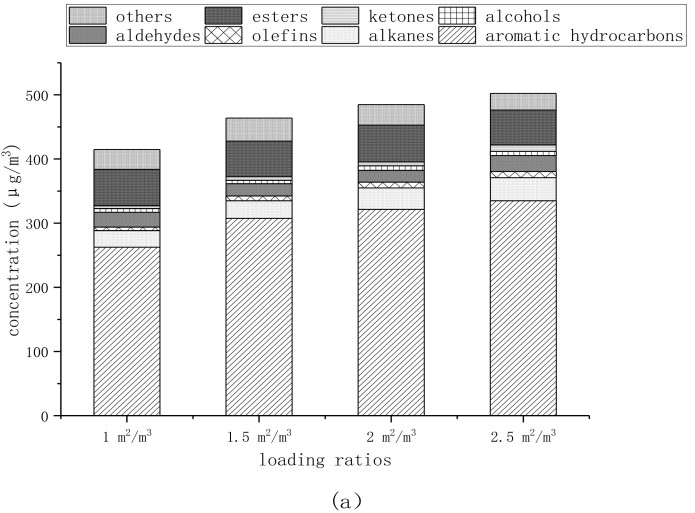

(a)

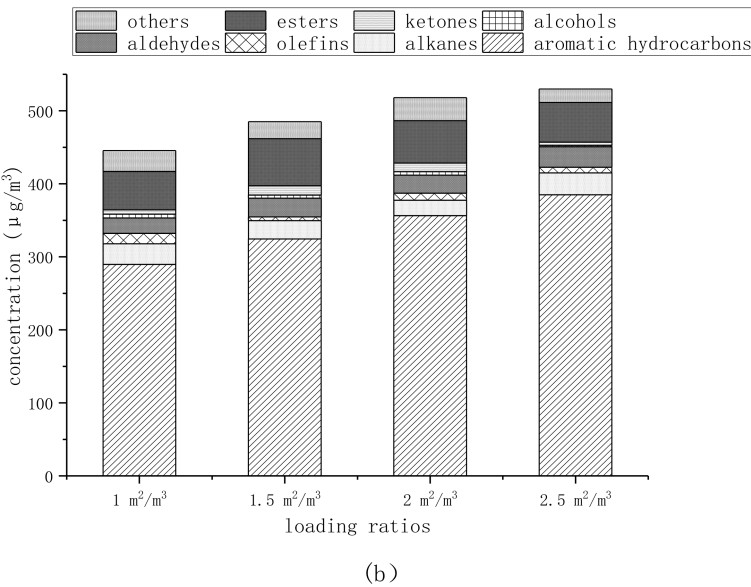

(b)

**Figure 4.** *Cont.*

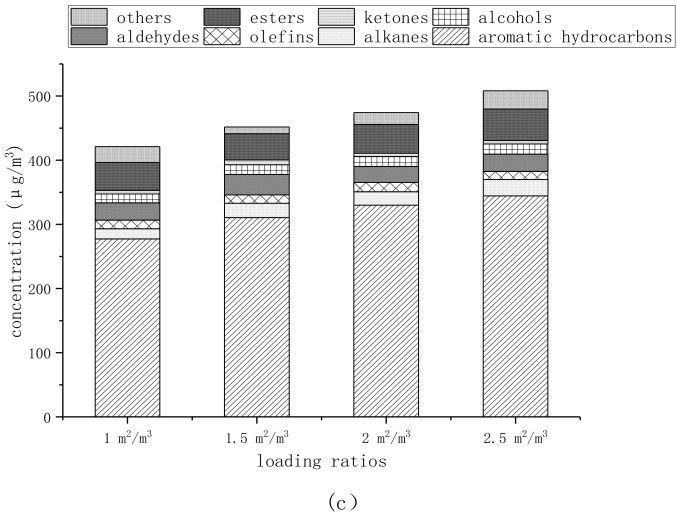

(c)

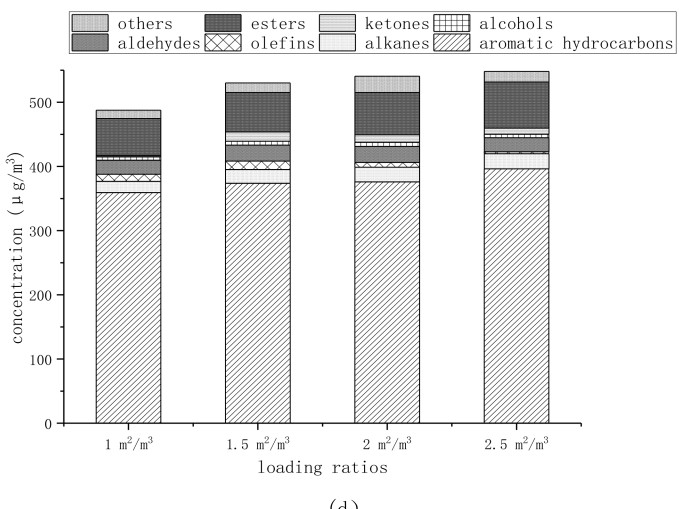

(d)

**Figure 4.** The VOC-components released from PVC-P (**a**), MI-P (**b**), UF-P (**c**) and WP-P (**d**) at 30 h.

## 4. Conclusions

The VOCs released from the four plywood types showed an upward trend in relation to the increase of time in an airtight environment. The values did not continually increase, but gradually reached saturation. The time required for the different types of laminated plywood to reach saturation was different. Overall, VOCs released by surface laminated plywood reached saturation faster, requiring only 12 h, whereas UF-P required 18 h. In terms of VOC concentration at saturation, the pollution degree of the three veneered plywood from lowest to highest is PVC-P, MI-P and WP-P in turn. Aromatic hydrocarbons account for the largest proportion which was more than half of TVOC concentration among VOC-components and was most obviously affected by the loading ratios. Veneered plywood can reduce the emission of aromatic hydrocarbons, but also increased the amount of other compounds released. Not all the growth characteristics of VOCs showed a logarithmic growth trend as expected. Due to the influence of laminate materials, the concentration of various VOC-components shows different characteristic curves at different loading ratios.

**Author Contributions:** J.S. designed the experiment, T.C. and C.X. did the experiment and wrote the manuscript, Q.W., H.D. and H.L. analyzed the data. All authors contributed to the discussion of the results and have read and approved the final manuscript.

**Funding:** This research was funded by the National Key Research and Development Program of China (Grant 2016 YFD 0600706).

**Acknowledgments:** This study was supported by the National Key Research and Development Program of China (Grant 2016 YFD 0600706).

**Conflicts of Interest:** The authors declare no conflict of interest.

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
