# Peer review of "Characteristics of VOCs Released from Plywood in Airtight Environments"

_forests, doi:10.3390/f10090709_

Round 1

Reviewer 1 Report

This is an excellent publication and I find it of very real interest in these days where VOC emissions are a major issue. I would suggest to the authors only a change to render it easier to a reader to understand the results: Figures 2 and 3 are difficult to read simply because in different variation of grey is dfficult to understand and see what is what. I suggest two solutions, to the choice of the authors:

Either do the figures in different colours so that one can know for instance in Figure 2 what VOC is what. Even better would be that Figure 2 be split in  4 figures if the authors leave the "pie-charts" grey as they are so that each pie-chart can be enlarged and more easy to see what VOC belongs to which sector of the pie-chart. Equally for Figure 3 please split it into 2 figures and enlarge the graphs.

Author Response

Dear Reviewer ,

Thank you for your suggestion, which has helped me a lot.

I have split and enlarged Figures 2 and 3 (now they are Figures 3 and 4) which now is clearer.

Reviewer 2 Report

This paper is interesting in measuring VOCs from surface laminated plywood with different materials. However,b there are some changes and revision is needed:

The term 'veneered' shoudl changed to 'surface laminated' for the whole manuscript. The 'panel are to volume ratio' should changed to 'loading ratio' with the same unit (m^2/m^3). One of then serious weakness of this work is the fact that the sample was not exposed to a dynamic air flow. So, the results are not compatible with a real condition in building which has a certain air exchange per hour, i.e., ventilation in building. It is highly recommend for the authors to compare the presented results with those at dynamic condition in future. One of the question is what are the components of other hydrocarbons over carbon-16 in this paper. I believe that they are natural VOCs, including terpens. It is highly recommended for the authors to analyze these components. Please refer to a reference (Journal of Adhesion Science and Technology, 2013, 2(5-6): 620–631) to compare the data. Other things are marked in the manuscript in yellow color.

Author Response

Dear reviewer, your suggestions are correct and very helpful to me.

1. I've replaced “veneered” with “surface laminated”, also the “panel area to volume ratios” has been replaced by “loading ratios”.

2. Thank you for your suggestion. The airtight experiment is only a part of my overall experiment. Before that, I did corresponding experiments in dynamic condition(the gas change rate is 1 h-1). The emission characteristics of VOCs from surface laminated plywood could be simulated by dynamic experiments under ideal ventilation conditions. While the airtight experiments could simulate characteristics of VOCs of an enclosed living environment under extreme conditions. Both experiments have guiding significance for the selection of panel materials in living environment under different ventilation conditions. At present, there are many researches on VOCs released in dynamic environment. Therefore, it is necessary to study the characteristics of VOCs released in airtight environment. I have added a part about comparing dynamic experiments with airtight experiments.

3. TVOC is sum of the concentrations of identified and unidentified volatile organic compounds eluting between and including n-hexane and n-hexadecane (ISO-16000).In this airtight, I am sorry that the components of other hydrocarbons over carbon-16 are not belong to my field of study.

4. The references you recommend are very helpful. I compared some data of VOC-Components and got a good correlation with my experiment. I have added a reference to this article to the manuscript.

Round 2

Reviewer 1 Report

I have no qualms in recommending that the paper be considered as is